# Arboreal Neural Network

**Wubin Yan** [1,2]  **Wei Ma** [2]  **Shixiang Wan** [1]  **Dongchen Li** [1]  **Shaoshun Kang** [1]  **Qing Yang** [1]  **Dongliang Xu** [1]

## Abstract

Connectionist models and symbolic models have long embodied two divergent paradigms: the former excel at differentiable representation learning yet struggle with transparency, while the latter deliver explicit rule-based reasoning but resist gradient-based optimization. We introduce Arboreal Neural Networks (ArbNN), a neural-symbolic framework that unifies these paradigms both computationally and conceptually. At the design level, ArbNN departs fundamentally from prior neuralized-tree models through a depth-aware routing mechanism and a topology-informed softmax aggregation, which together enable one-shot full-path gradient propagation and consequently achieving rapid and well-conditioned optimization dynamics. At the conceptual level, ArbNN reveals that decision-tree branching and self-attention routing are two realizations of the same conditional computation primitive. We prove a structural isomorphism between a decision tree and a single-query attention head, enabling a differentiable architecture that faithfully preserves symbolic decision logic. The defining property of ArbNN is Bidirectional Fidelity, ensuring that the neural module can be compiled from—and losslessly decompiled back into—a symbolic tree, yielding both ordering consistency in ranking behavior and explicit, auditable interpretability via reconstructed if-else rules. ArbNN further supports GBDT-based initialization, allowing it to inherit strong inductive biases and integrate seamlessly with existing production workflows. Empirically, ArbNN achieves state-of-the-art performance on various public tabular benchmarks and delivers consistent gains under temporal distribution shift in large-scale industrial credit-risk systems. To support realistic evaluation, we additionally construct TabCredit, a feature-rich, temporally partitioned dataset built from millions of real-world loan applications. Together, these results demonstrate that ArbNN forms a unified, reversible, and practically deployable bridge between symbolic reasoning and neural computation for high-stakes tabular domains.

## 1. Introduction

A prevailing challenge in unifying symbolic (Chen & Guestrin, 2016; Ke et al., 2017; Prokhorenkova et al., 2018) and neural paradigms (Lecun et al., 1998; Krizhevsky et al., 2012; He et al., 2016; Vaswani et al., 2017; Radford et al., 2018; Devlin et al., 2019) lies not merely in hybridizing tree-like components with neural modules, but in determining whether they share a common conditional computation primitive. Prior neural-symbolic approaches often soften discrete decision boundaries, approximate symbolic operators with differentiable surrogates, or impose post-hoc explanations on neural networks. While such techniques yield practical utility, they do not bridge the core conceptual divide: the lack of a principled and invertible correspondence between discrete symbolic structures and continuous neural representations. Recent work in modular and interpretable neural systems further highlights this difficulty. Typical studies on Weight-Sparse Transformers (Gao et al., 2026) and Modular Manifolds (Bernstein, 2025) demonstrate that learning sparse, modular, or symbolic structure *from scratch* remains intrinsically challenging. Neural activations tend to remain entangled and polysemantic, even under strong regularization, and enforcing routing, modularity, or discreteness during training is often unstable and hardly yields audit-grade interpretability.

We introduce Arboreal Neural Networks (ArbNN), a neural architecture that provides a bidirectional correspondence between decision trees and self-attention. A decision tree can be compiled into a differentiable module that is structurally homomorphic to a single-query attention head, and the resulting module can be decompiled back into an equivalent symbolic tree with no loss of interpretability. This establishes attention-based routing and symbolic branching as two forms of the same conditional computation primitive,

[1]Du Xiaoman Technology (Beijing) Co., Ltd., Beijing, China [2]Beijing University of Technology, Beijing, China. Correspondence to: Wei Ma <mawei@bjut.edu.cn>.

*Proceedings of the 43rd International Conference on Machine Learning*, Seoul, South Korea. PMLR 306, 2026. Copyright 2026 by the author(s).

enabling ArbNN to combine global, auditably symbolic reasoning with end-to-end trainability and competitive function approximation. ArbNN employs a more principled and informative initialization that uses a pretrained GBDTs as a structural sparsity template, rather than relying on a purely random sparse topology as in weight-sparse Transformer regimes (Gao et al., 2026). This design provides a meaningful and domain-aligned structural prior, following the broader shift toward fixed-structure models with differentiable parameter refinement. More realistically for real-world deployments, it also inherits the most widely used tree models, avoiding migration cost or performance risk and providing a stable foundation for fine-tuning.

We also construct an industrial dataset. As is well-known, most academic tabular benchmarks adopt random splits to construct their datasets, with the aim of evaluating generalization under the assumption of Independent and Identically Distributed (I.I.D.) data. However, these benchmarks often overlook the temporal drift and strong time dependencies inherent in real-world applications. Additionally, industrial datasets are typically created through complex acquisition processes and extensive feature engineering, leading to numerous informative variables. Despite their value, such datasets are rarely publicly available (Rubachev et al., 2025). In this paper, we present **TabCredit**, a large-scale, feature-rich credit risk dataset derived from real-world loan applications and organized with temporal splits. TabCredit enables benchmarking under realistic conditions of temporal drift and complex feature engineering—two critical aspects that are largely absent from existing public datasets.

Our main contributions are summarized as follows:

- We present **Arboreal Neural Networks (ArbNN)**, a neural–symbolic framework built around *bidirectional fidelity*: the model can be compiled from and decompiled back into a symbolic tree while preserving ordering-consistent ranking behavior and explicit, auditable interpretability. ArbNN further unifies symbolic decision-tree reasoning with connectionist representation learning in a single differentiable architecture.

- We introduce **ArborCell**, the differentiable computation unit of ArbNN. ArborCell employs a depth-aware routing mechanism together with a topology-informed softmax aggregation, enabling one-shot full-path computation for efficient training. We additionally prove that ArborCell is *structurally homomorphic* to a single-query self-attention head, establishing the formal correspondence between symbolic branching and neural routing.

- Our model achieves State-of-the-Art (SOTA) performance across various tabular prediction datasets, outperforming both traditional ensemble trees and existing neural models. In addition, it has been successfully deployed in real-world commercial risk control scenarios involving large-scale credit flow, demonstrating its practical efficacy.

- We introduce **TabCredit**, a large-scale and feature-rich dataset for credit risk assessment, constructed from real-world loan applications. The dataset comprises 2 million samples with 100 anonymized, engineered features and adheres to strict in-time and Out-of-Time (OOT) splits.

**Conflict of Interest Disclosure.** The authors are employed by Du Xiaoman Technology (Beijing) Co., Ltd., which provided the computational resources and proprietary dataset (TabCredit) for this research. Additionally, the proposed ArbNN architecture is currently deployed within the company's live commercial systems.

## 2. Related Work

### 2.1. Traditional Tree-based Models

Traditional tree-based models, including XGBoost (Chen & Guestrin, 2016), LightGBM (Ke et al., 2017), and Cat-Boost (Prokhorenkova et al., 2018), remain the foundational paradigm for tabular data modeling. By constructing hierarchical feature routing systems based on split points determined from statistical criteria (e.g., information gain), they naturally align with the localized dependencies and additive interactions common in such data. Furthermore, recent empirical studies, such as the comprehensive benchmarking by (Holzmüller et al., 2024), emphasize that these GBDT frameworks can achieve highly competitive performance across diverse tabular tasks simply through the application of robust, carefully pre-tuned hyperparameter configurations.

Despite their empirical success, traditional GBDTs are fundamentally limited by their greedy optimization based on global statistical aggregates, which frequently causes them to underfit rare or challenging minority instances. To address this, ArbNN renders the tree architecture fully end-to-end differentiable. By freezing the physical topology and employing batch-wise gradient descent to globally co-optimize all split thresholds and leaf values, ArbNN effectively captures these underrepresented patterns, directly yielding superior performance.

### 2.2. Differentiable Neural Architectures

End-to-end differentiable architectures have been widely explored for tabular data. TabTransformer (Huang et al., 2020) is an early work of applying self-attention to embeddings of categorical features within tabular data, enabling context-aware feature encoding. FT-Transformer (Gorish-

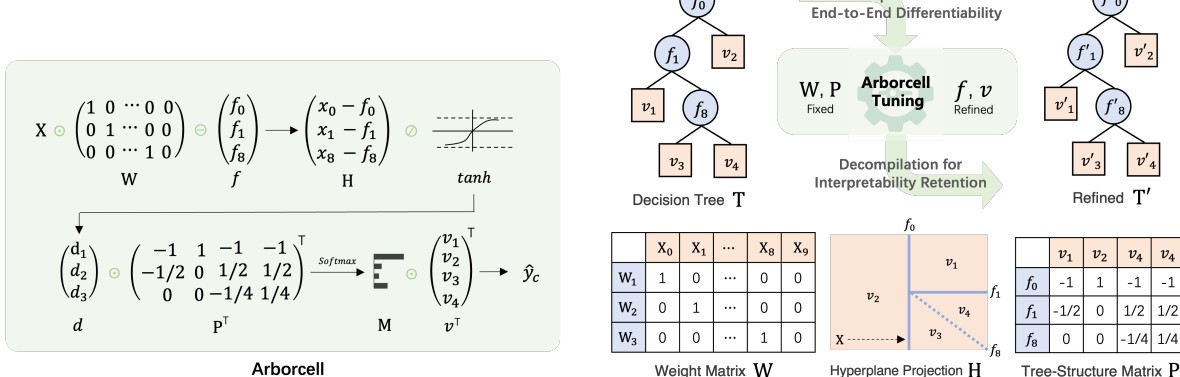

*Figure 1.* Arboreal Neural Network transforms a decision tree $\mathbf{T}$ into an ArborCell, enabling **end-to-end differentiability** while retaining the symbolic structure for **interpretability**. The left-green panel illustrates the compilation of tree $\mathbf{T}$ (via the Algorithm in Appendix A into its ArborCell representation. Matrix $\mathbf{W}$ encodes feature–node assignments, and matrix $\mathbf{P}$ encodes the tree topology. Together with the bias vector $f$ (split thresholds), these define the hyperplane projections that relate input $\mathbf{X}$ to the internal decision nodes. After training, the ArborCell can be decompiled back into a refined tree $\mathbf{T}'$, where $(\mathbf{W}, \mathbf{P})$ remain fixed and only $(f, v)$ are fine-tuned, thereby improving predictive performance while preserving the original tree structure.

niy et al., 2021) expanded on TabTransformer by jointly processing numerical and categorical features through a unified Transformer block. TFWT (Zhang et al., 2024) introduced dynamic feature weighting within Transformer layers to highlight context-dependent variable importance. SwitchTab (Wu et al., 2024) employed an asymmetric encoder-decoder structure to enhance feature representations and interpretability. Trompt (Chen et al., 2024b) utilized prompt-based learning to separate global schema information from instance-specific noise. TabM (Gorishniy et al., 2025) achieved ensemble benefits for tabular deep learning by combining a shared MLP backbone with lightweight adapters for parameter-efficient diversity. TabPFNv2 (Hollmann et al., 2025) is a pre-trained tabular foundation model that, by learning to perform Bayesian-style inference from diverse synthetic priors, produces accurate zero-shot predictions on small-to-medium tabular datasets while delivering orders-of-magnitude faster inference than tuned AutoML baselines.

Despite their adaptability in handling diverse tabular data, differentiable neural tabular models often struggle to consistently outperform tree ensembles across many tasks—particularly in scenarios with limited data or heavily engineered features (Rubachev et al., 2025). Moreover, these models lack the intuitive interpretability inherent to tree-based models.

### 2.3. Hybrid Neural-Tree Architectures

Hybrid neural-tree architectures differ mainly in how decision-tree logic is integrated with neural computation. Early approaches implement *independent soft trees*, where the predictive model itself is structured as a differentiable ensemble. These methods replace hard if-else splits with

*soft gating* (sigmoid or softmax) and compute predictions as mixtures over leaves via *path probabilities*. Representative examples include Neural Decision Forests (Kontschieder et al., 2015) and Soft Decision Trees (Frosst & Hinton, 2017), which use per-node binary soft splits (sigmoid-based left/right routing); Neural Random Forests (Biau et al., 2019) implement a depth-agnostic routing mechanism in which each internal node computes a signed matrix product over input features, producing a single-leaf activation that differentiably mimics the hard path selection of classical decision trees. NODE (Popov et al., 2020), which performs layerwise oblivious soft splits and GRANDE (Marton et al., 2024), combines softmax-based feature selection with sigmoid-based thresholding to stabilize gradient flow in deep tree ensembles. Another direction embeds tree-based reasoning within neural networks as specialized modules, such as the Adaptive Neural Trees (Tanno et al., 2019), which insert learnable neural Transformers along paths and dynamically grow or prune the tree to enable adaptive structure learning; Tree Ensemble Layer (TEL) (Hazimeh et al., 2020), which integrates differentiable tree ensembles as conditional-computation layers; Adaptive Neural Trees (Tanno et al., 2019), which insert learnable neural Transformers along paths and dynamically grow or prune the tree to enable adaptive structure learning; Net-DNF (Katzir et al., 2021), which encodes symbolic decision logic in disjunctive normal form as neural layers, preserving rule-based reasoning; and NCART (Luo & Xu, 2024), which incorporates differentiable oblivious trees into residual networks to improve both efficiency and interpretability. ArbNN overcomes the optimization instability and semantic opacity of prior soft-tree models by replacing recursive gating with a structurally grounded, one-shot matrix formulation. By integrating depth-aware routing with topology-informed

| Model | Classification ↑ | | | | | | | Regression ↓ | | | |
|---|---|---|---|---|---|---|---|---|---|---|---|
| | MA | H16 | CR | JA | DI | MI | CA | YP | HO | CH | ME |
| MLP (Haykin, 1994) | 0.938(1) | 0.940(2) | 0.830(2) | 0.869(1) | 0.650(3) | 0.978(1) | 0.953(2) | 0.881(1) | 0.435(1) | 0.467(2) | 0.140(1) |
| ResNet (He et al., 2016) | 0.938(2) | 0.946(1) | 0.828(3) | 0.871(2) | 0.649(2) | 0.983(1) | 0.960(2) | 0.875(1) | 0.434(2) | 0.468(1) | 0.147(2) |
| FTTransformerBucket (Gorishniy et al., 2021) | 0.939(5) | 0.950(4) | 0.857(6) | 0.866(5) | 0.649(7) | 0.986(4) | 0.960(5) | 0.885(4) | 0.441(6) | 0.468(5) | 0.147(4) |
| FTTransformer (Gorishniy et al., 2021) | 0.936(2) | 0.949(1) | 0.828(2) | 0.871(1) | 0.652(2) | 0.981(2) | 0.959(1) | 0.896(2) | 0.445(1) | 0.461(2) | 0.141(1) |
| TabNet (Arik & Pfister, 2021) | 0.937(3) | 0.936(2) | 0.828(2) | 0.862(3) | 0.650(1) | 0.977(2) | 0.944(3) | 0.885(2) | 0.451(3) | 0.544(2) | 0.140(1) |
| TabTransformer (Huang et al., 2020) | 0.918(2) | 0.937(3) | 0.816(1) | 0.867(2) | 0.642(2) | 0.980(1) | 0.928(2) | 0.886(1) | 0.539(2) | 0.545(3) | 0.141(1) |
| Trompt (Chen et al., 2024b) | 0.942(1) | 0.952(1) | 0.829(2) | 0.882(1) | 0.652(1) | 0.982(2) | 0.964(1) | 0.874(2) | 0.418(1) | 0.424(2) | 0.140(1) |
| GRANDE (Marton et al., 2024) | 0.939(2) | 0.945(2) | 0.833(1) | 0.871(2) | 0.653(1) | 0.982(1) | 0.959(2) | 0.883(1) | 0.432(2) | 0.460(1) | 0.141(2) |
| ExcelFormer (Chen et al., 2024a) | 0.939(1) | 0.950(1) | 0.833(1) | 0.879(2) | 0.651(2) | 0.982(1) | 0.969(1) | 0.890(2) | 0.411(1) | 0.431(2) | 0.142(1) |
| SwitchTab (Wu et al., 2024) | 0.939(2) | 0.948(2) | 0.828(2) | 0.872(1) | 0.651(1) | 0.981(1) | 0.961(2) | 0.880(1) | 0.430(2) | 0.452(1) | 0.144(2) |
| TabPFNv2 (Hollmann et al., 2025) | 0.939(1) | 0.946(1) | 0.854(2) | 0.873(1) | 0.638(2) | 0.980(2) | 0.967(1) | 0.888(1) | 0.400(1) | 0.452(2) | 0.143(1) |
| TabM (Gorishniy et al., 2025) | 0.943(1) | 0.947(2) | 0.843(1) | 0.875(2) | 0.651(1) | 0.982(1) | 0.963(2) | 0.877(2) | 0.396(1) | 0.435(1) | 0.142(1) |
| CatBoost (Prokhorenkova et al., 2018) | 0.924(2) | 0.948(1) | 0.861(2) | 0.863(1) | 0.649(2) | 0.986(1) | 0.963(1) | 0.876(1) | 0.443(2) | 0.439(1) | 0.145(1) |
| LightGBM (Ke et al., 2017) | 0.943(1) | 0.951(1) | 0.862(2) | 0.881(2) | 0.652(1) | 0.986(0) | 0.972(1) | 0.863(2) | 0.384(1) | **0.395(1)** | 0.141(1) |
| XGBoost (Chen & Guestrin, 2016) | 0.947(1) | 0.951(0) | 0.862(1) | 0.868(1) | 0.653(2) | 0.986(1) | 0.966(1) | 0.857(1) | 0.401(2) | 0.418(1) | 0.140(0) |
| ArbNN | **0.974(1)** | **0.987(1)** | **0.884(1)** | **0.923(2)** | **0.657(1)** | **0.992(0)** | **0.986(1)** | **0.832(1)** | **0.359(1)** | 0.406(1) | **0.130(1)** |

*Table 1.* Comparisons on public datasets, with the highest performance highlighted in **bold**. Values are reported as $mean(std \times 10^3)$, where the parentheses denote the standard deviation corresponding to the last significant digit(s) over five independent runs. The evaluation metrics are AUC-ROC for classification and RMSE for regression.

softmax, it achieves stable end-to-end training. Crucially, ArbNN guarantees Bidirectional Fidelity, where soft neural inference is ranking-equivalent to the hard symbolic decisions (matching up to the 5th decimal place). This property allows for lossless decompilation, effectively unifying connectionist representation learning with symbolic reasoning under a shared primitive.

## 3. Method

### 3.1. Arboreal Neural Network

Arboreal Neural Network is a wide, shallow and differentiable neural architecture that reinterprets the discrete logic of tree-based models. Each component, called an Arbor-Cell, functions similarly to a tree in tree-based models. The overall prediction of the ArbNN model is obtained by aggregating the logits from its individual ArborCells:

$$\hat{y}_{\text{ArbNN}} = \sum_{c=1}^{C} \hat{y}_c \quad (1)$$

where $C$ denotes the total number of ArborCells in the ArbNN model. The ArbNN model is initially configured using a pretrained decision tree ensemble (e.g., XGBoost), which provides critical elements, including split conditions, leaf node weights, and structural topology.

### 3.2. Sparse Neural Tree: ArborCell

As illustrated in Figure 1, each ArborCell processes a sample $\mathbf{X} \in \mathbb{R}^t$ and produces a prediction $\hat{y}_c$. This process emulates tree prediction logic through a series of neural computations, including matrix multiplication and activation functions. The necessary matrices are derived from a decision tree, denoted as $\mathbf{T}$, using Algorithm in Appendix A. Specifically, we extract the sparse weight matrix $\mathbf{W}$ that

indicates feature-node relationships, a bias vector $f$ that represents split points, a tree-structure matrix $\mathbf{P}$ that reflects the hierarchical subspace organization, and leaf node weights $v$.

**Weight Matrix** $\mathbf{W} \in \mathbb{R}^{n_i \times t}$ is a sparse one-hot matrix, where each row corresponds to an internal node, and the non-zero entry in row $i$ indicates the index of the feature selected by the $i$-th internal node from a set of $t$ candidate features. $n_i$ is the total number of inner nodes.

**Tree-structure Matrix** $\mathbf{P} \in \mathbb{R}^{n_i \times n_j}$ captures the tree's structure. Each entry $p_{ij}$ indicates whether the leaf node $j$ is reachable from the internal node $i$ and on which side of the hyperplane the leaf lies. Specifically, for each pair of nodes, the value of $p_{ij}$ is defined as:

$$p_{ij} = \begin{cases} \left(\frac{1}{2}\right)^{dep_j - 1}, & \text{if node } j \in \text{Desc}_{\text{L}}(i) \\ -\left(\frac{1}{2}\right)^{dep_j - 1}, & \text{if node } j \in \text{Desc}_{\text{R}}(i) \\ 0, & \text{if node } j \notin \text{Desc}(i) \end{cases} \quad (2)$$

where $dep_j$ represents the number of edges from the root to leaf node $j$ and $\text{Desc}_{\text{L}}(i)$ and $\text{Desc}_{\text{R}}(i)$ denote the sets of leaf nodes that lie on the left or right subtrees of node $i$, respectively. The sign indicates whether a leaf is on the left or right of the hyperplane ($+1$ for left and $-1$ for right) defined by node $i$, and the magnitude is weighted by $(1/2)^{dep_j - 1}$. A value of $0$ indicates that the leaf is not reachable from node $i$. We adopt an exponentially decaying weighting design for the matrix $\mathbf{P}$, specifically utilizing a base of $1/2$. This specific base is derived directly from the topological prior of a binary tree: in a perfectly balanced structure, the prior probability of a sample routing to either the left or right subtree halves at each depth level. By embedding this $1/2$ decay into the matrix forward pass, we faithfully emulate this hierarchical subspace selection in a fully end-to-end learnable manner, while simultaneously mitigating the gradient van-

| Model | MOB2 | MOB3 | MOB4 | MOB5 | MOB6 | MOB7 | MOB8 | MOB9 | MOB10 | MOB11 | MOB12 |
|---|---|---|---|---|---|---|---|---|---|---|---|
| MLP (Haykin, 1994) | 54.73 | 52.19 | 50.35 | 48.92 | 47.24 | 45.68 | 43.91 | 42.49 | 41.26 | 39.99 | 38.82 |
| ResNet (He et al., 2016) | 55.39 | 52.89 | 50.94 | 49.39 | 47.57 | 45.90 | 44.06 | 42.70 | 41.48 | 40.20 | 38.96 |
| FTTransformer (Gorishniy et al., 2021) | 55.77 | 53.36 | 51.32 | 49.57 | 47.70 | 46.03 | 44.12 | 42.67 | 41.38 | 40.04 | 38.80 |
| TabTransformer (Huang et al., 2020) | 51.25 | 48.78 | 47.11 | 45.40 | 43.67 | 42.11 | 40.51 | 39.22 | 38.08 | 36.91 | 35.74 |
| FTTransformerBucket (Gorishniy et al., 2021) | 55.80 | 53.42 | 51.35 | 49.61 | 47.75 | 46.09 | 44.16 | 42.71 | 41.43 | 40.07 | 38.85 |
| Trompt (Chen et al., 2024b) | 55.54 | 52.88 | 50.99 | 49.22 | 47.34 | 45.68 | 43.85 | 42.37 | 41.05 | 39.72 | 38.46 |
| GRANDE (Marton et al., 2024) | 55.49 | 52.80 | 50.91 | 49.13 | 47.24 | 45.59 | 43.72 | 42.22 | 40.89 | 39.64 | 38.38 |
| ExcelFormer (Chen et al., 2024a) | 53.80 | 51.69 | 49.74 | 48.36 | 46.77 | 45.34 | 43.71 | 42.40 | 41.22 | 40.01 | 38.79 |
| SwitchTab (Wu et al., 2024) | 55.57 | 53.12 | 51.11 | 49.46 | 47.61 | 45.91 | 44.02 | 42.53 | 41.21 | 39.85 | 38.67 |
| TabPFNv2 (Hollmann et al., 2025) | 28.42 | 29.20 | 29.46 | 29.83 | 29.66 | 29.71 | 28.88 | 30.11 | 31.17 | 29.85 | 30.61 |
| TabM (Gorishniy et al., 2025) | 55.84 | 53.47 | 51.43 | 49.57 | 47.84 | 46.19 | 44.26 | 42.75 | 41.49 | 40.13 | 39.01 |
| XGBoost (Chen & Guestrin, 2016) | 56.52 | 53.92 | 51.90 | 50.13 | 48.19 | 46.40 | 44.50 | 43.03 | 41.72 | 40.40 | 39.13 |
| LightGBM (Ke et al., 2017) | 56.31 | 53.72 | 51.65 | 49.90 | 48.01 | 46.18 | 44.35 | 42.78 | 41.55 | 40.21 | 38.95 |
| ArbNN | **57.14** | **54.13** | **52.08** | **50.33** | **48.43** | **46.65** | **44.76** | **43.28** | **41.95** | **40.60** | **39.32** |

*Table 2.* KS (%) across M2+@MOBN in vintage analysis on the TabCredit Dataset for tracking delinquency trends (M2+: delinquency beyond 31 days; MOBN: months-on-book). The best values are in bold.

ishing and exploding issues inherent to deeply skewed tree structures during backpropagation.

### 3.3. Feed Forward Computation in ArborCell

**Hyperplane Projection H**: Each internal node defines a hyperplane in the feature space, represented by a weight-bias pair $(\mathbf{W}_i, f_i)$. To quantify the relationship of $\mathbf{X}$ to this hyperplane, we define the *hyperplane projection*:

$$\mathbf{H}_i = \mathbf{W}_i \mathbf{X} - f_i \tag{3}$$

Here, $\mathbf{H}_i$ represents the signed distances of $\mathbf{X}$ to the hyperplane. By evaluating $\mathbf{X}$ against $n_i$ learned hyperplanes and form $\mathbf{H} \in \mathbb{R}^{n_i \times 1}$.

**Decision Vector d**: The hyperplane projections are passed through a scaled hyperbolic tangent function to obtain the decision vector $\mathbf{d} \in \mathbb{R}^{n_i}$:

$$\mathbf{d}_i = \tanh(\tau_1 \cdot \mathbf{H}_i) \tag{4}$$

where $\tau_1 > 0$ is a temperature parameter that modulates the sharpness of the activation function. A higher $\tau_1$ pushes $\mathbf{d}_i$ closer to the extremal values $-1$ and $+1$, thereby approximating the hard binary decisions of traditional trees.

In this way, each element $\mathbf{d}_i$ reflects the direction and confidence of the input sample with respect to the $i$-th decision boundary: $d_i \approx +1$ indicates a confident right-child traversal, $d_i \approx -1$ indicates a left-child preference, and intermediate values arise when $\tau_1$ is small or the input lies near the hyperplane. As a whole, the vector $\mathbf{d}$ serves as a differentiable analogue of the discrete split routing in decision trees, encoding soft decisions over internal nodes.

**Subspace Affinity Vector M**: The decision vector $\mathbf{d} \in \mathbb{R}^{n_i}$ is linearly transformed by the tree-structure matrix $\mathbf{P} \in \mathbb{R}^{n_i \times n_j}$ to obtain the subspace affinity vector:

$$\mathbf{M} = \mathbf{P}^\top \mathbf{d} \in \mathbb{R}^{n_j} \tag{5}$$

where $n_j$ denotes the number of leaf nodes. Each element $\mathbf{M}_j$ reflects the aggregated directional alignment of the input with respect to the internal nodes along the path to leaf node $j$, incorporating both routing consistency and hierarchical depth weighting. This process can be viewed as a soft generalization of decision path traversal in conventional trees.

To obtain a normalized distribution over leaf subspaces, the subspace affinity vector $\mathbf{M}$ is passed through a softmax function: $\boldsymbol{\alpha} = \text{Softmax}(\tau_2 \cdot \mathbf{M})$, where the temperature parameter $\tau_2 > 0$ controls the peakedness of the distribution. The resulting vector $\boldsymbol{\alpha} \in \mathbb{R}^{n_j}$ encodes a probabilistic allocation of the input across all leaf regions, enabling smooth, differentiable transitions between them.

**ArborCell Prediction $\hat{y}_c$**: Let $\mathbf{v} \in \mathbb{R}^{n_j}$ denote the prediction values associated with leaf nodes. The final output of the ArborCell is given by the expectation over leaf predictions:

$$\hat{y}_c = \sum_{j=1}^{n_j} \alpha_j v_j \tag{6}$$

This output represents a soft aggregation of all leaf values, weighted by how consistently the input aligns with the decision paths encoded by the internal node activations and their hierarchical routing.

### 3.4. Structural Homomorphism to Self-Attention

We show that the computation of an *ArborCell*—the differentiable counterpart of a decision tree unit—is structurally homomorphic to a single-query self-attention head. We retain the notation from the Tree Parsing Algorithm (Appendix 1).

**Theorem.** An ArborCell is structurally homomorphic to a single-query self-attention head in the following sense.

*(1)* Each internal node $i \in \{1, \ldots, n_i\}$ contributes one coordinate to the decision vector $\mathbf{d} \in \mathbb{R}^{n_i}$, i.e., one axis of

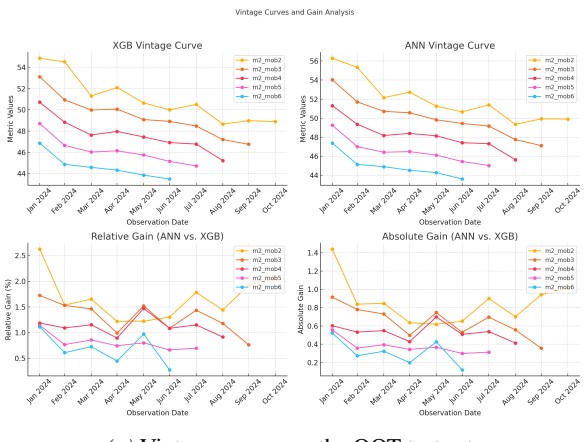

*(a)* Vintage curves on the OOT test set.

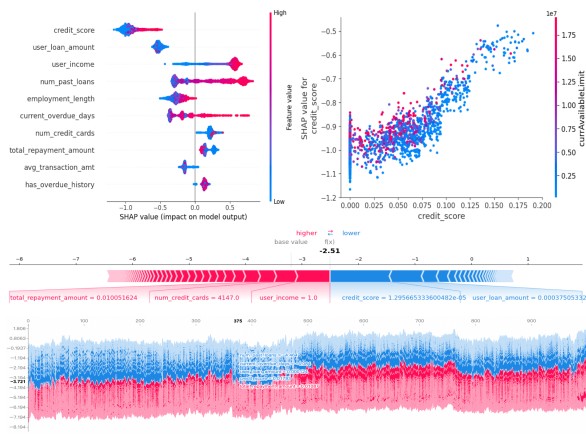

*(b)* TreeSHAP interpretability on TabCredit.

*Figure 2.* Performance and interpretability analysis of ArbNN.

the query embedding space. Hence the number of internal nodes $n_i$ equals the attention *embedding dimension*.

*(2)* Each leaf $\ell \in \{1, \ldots, n_j\}$ is associated with (a) a path encoding given by the $\ell$-th column of the tree-structure matrix $\mathbf{P} \in \mathbb{R}^{n_i \times n_j}$ and (b) a scalar leaf value $v_\ell$ from $\mathbf{v} \in \mathbb{R}^{n_j}$. Thus the number of leaves $n_j$ equals the *sequence length* (number of tokens).

Under the identification

$$\mathbf{Q}_{(\text{CLS})} = \mathbf{d}^\top \in \mathbb{R}^{1 \times n_i},$$
$$\mathbf{K}_{(\text{tokens})} = \mathbf{P}^\top \in \mathbb{R}^{n_j \times n_i}, \quad (7)$$
$$\mathbf{V}_{(\text{tokens})} = \mathbf{v} \in \mathbb{R}^{n_j \times 1}$$

The ArborCell output matches the single-query attention form:

$$\text{ArborCell}(\mathbf{d}, \mathbf{P}, \mathbf{v}) = \text{Attention}\big(\mathbf{Q}_{(\text{CLS})}, \mathbf{K}_{(\text{tokens})}, \mathbf{V}_{(\text{tokens})}\big) \quad (8)$$

**Proof.**

*(1) Subspace affinity.* By the ArborCell definition (Sec. 3.2), the subspace affinity vector is $\mathbf{M} = \mathbf{P}^\top \mathbf{d} \in \mathbb{R}^{n_j}$. Using the above identification, this is equivalently

$$\mathbf{M} = \mathbf{K}_{(\text{tokens})} \, \mathbf{Q}_{(\text{CLS})}^\top \in \mathbb{R}^{n_j \times 1} \quad (9)$$

which is precisely the key-query affinity in self-attention.

*(2) Normalization.* Routing weights over leaves are obtained by normalizing the affinities:

$$\boldsymbol{\alpha} = \text{softmax}\big(\tau_2 \mathbf{K}_{(\text{tokens})} \, \mathbf{Q}_{(\text{CLS})}^\top\big) \in \mathbb{R}^{n_j} \quad (10)$$

mirroring attention-weight normalization. Standard $1/\sqrt{n_i}$ scaling can be incorporated into $\tau_2$ without affecting the argument.

*(3) Aggregation.* The ArborCell prediction aggregates leaf values under these routing weights:

$$\hat{y}_c = \boldsymbol{\alpha}^\top \mathbf{v} = \text{softmax}\big(\tau_2 \mathbf{K}_{(\text{tokens})} \, \mathbf{Q}_{(\text{CLS})}^\top\big) \mathbf{V}_{(\text{tokens})} \quad (11)$$

**Boundary Clarification.** We emphasize that the established isomorphism serves as a theoretical bridge to interpret sparse self-attention mechanisms from a tree-based perspective. While current research often prioritizes empirical outcomes over model weight distributions, we argue both are equally crucial for interpretability. Nevertheless, this mapping remains strictly structural. It provides no practical contribution to ArbNN's empirical performance, demonstrating topological alignment rather than functional equivalence to broader Transformer usage or learned structures.

## 4. Experiments

### 4.1. Experimental Setup

**Datasets:** We evaluated ArbNN using both public benchmarks and a self-constructed industrial dataset. The public datasets, obtained from the *PyTorch Frame* project (Hu et al., 2024), include Magic Gamma Telescope (MA), California Housing (CH), House 16H (H16), Jannis (JA), Diabetes130US (DI), MiniBooNE (MI), YProp 4.1 (YP), California (CA), Medical Charges (ME), Houses (HO), and Credit (CR). We used the official splits provided by *PyTorch Frame*. For datasets lacking standard splits, we randomly partitioned the data into training, validation, and testing sets, ensuring these identical splits were consistently applied across all model experiments for a fair comparison. These benchmarks typically contain tens of thousands of samples but have limited representation of negative classes. Additionally, their relatively small test sets constrain the statistical reliability of model performance comparisons.

To address the limitations of existing small-scale and static academic datasets, we introduce **TabCredit**, a large-scale commercial credit dataset comprising roughly 1 million loan application records per month, each with 100 anonymized and carefully engineered features. TabCredit is temporally partitioned into a training set (September 2023) and an OOT test set (November 2023), enabling rigorous evaluation under temporal drift. Each record includes future risk outcomes from two to twelve months-on-book (MOB2–MOB12), with default rates that naturally evolve over time, offering a realistic benchmark for risk modeling. Unlike many curated academic datasets, TabCredit preserves its native distribution, thereby reflecting real-world challenges of scale, drift, and heterogeneity. The dataset will be released upon publication to support further research. More details about TabCredit are provided in Appendix B.

**Data preprocessing:** To ensure each modeling paradigm is evaluated under its respective optimal conditions, we applied paradigm-specific preprocessing. For GBDTs and ArbNN, we retained the native scale of the datasets, as their split-routing mechanisms are inherently scale-invariant. Conversely, all other neural network baselines were trained on normalized inputs to ensure effective optimization. Regarding feature processing, missing numerical values were imputed using the minimum observed value per feature. Categorical features were processed via standard one-hot encoding; for ArbNN, this explicitly inherits XGBoost's default approach, translating categorical splits into binary decisions to structurally support the formulation of our differentiable routing matrices.

**Implementation and Training Details:** To ensure fairness in all comparative experiments, we applied consistent hyperparameter tuning procedures across all methods. In particular, tree-based baselines such as XGBoost, as well as all neural network baselines, were optimized using Optuna with identical search space constraints, consistent with the *PyTorch Frame* benchmark framework (Hu et al., 2024). For the proposed ArbNN, model parameters were initialized using pretrained XGBoost models from the same comparison group. For classification tasks, the final prediction was computed by summing the logits produced by all tree neurons and applying a sigmoid activation. For regression tasks, the logits were directly summed and a constant bias of 0.5 was added to improve initial convergence. We used the AdamW optimizer (Loshchilov & Hutter, 2017) with a learning rate of $1 \times 10^{-4}$, a batch size of 2048, and a weight decay of 0.02. The learning rate followed a two-phase schedule: a linear warmup over the first two epochs, followed by linear decay. All experiments were conducted using Python 3.8 and PyTorch on NVIDIA Tesla A100 GPUs. To emulate deterministic tree behavior while preserving differentiability, we used a fixed temperature parameter $\tau_1$ and $\tau_2$ to ensure that ArborCell units effectively approximate hard-split logic.

We empirically validated this setting through ablation experiments over $\tau_1, \tau_2 \in \{5, 10, 50, 100, 200\}$ and observed that $\tau_1 = 100$ and $\tau_2 = 10$ offered the optimal trade-off between accuracy and smoothness.

**Evaluation Metrics:** To evaluate the performance of the proposed ArbNN, we used AUC-ROC for classification tasks and RMSE for regression tasks on public datasets, reflecting discriminative power and predictive accuracy, respectively. For our large-scale commercial credit-risk dataset, we conducted a vintage analysis using the Kolmogorov-Smirnov (KS) statistic on M2+@MOB2 through M2+@MOB12, where M2+ denotes delinquency beyond 31 days and MOBN represents months-on-book. We also report Lift to assess the model's ranking quality. Furthermore, given the highly imbalanced nature of credit risk evaluation, we additionally compute the Area Under the Precision-Recall Curve (AUPRC) as a cost-sensitive ranking metric, alongside Brier Score and Expected Calibration Error (ECE) to rigorously assess probability calibration.

| Metric | Opt. | XGB | ArbNN | Impr. |
|---|---|---|---|---|
| AUC-ROC | ↑ | 0.8188 | **0.8194** | +0.07% |
| Lift | ↑ | 1.8440 | **1.8442** | +0.01% |
| AUPRC | ↑ | 0.1166 | **0.1168** | +0.17% |
| Brier | ↓ | 0.0246 | **0.0241** | +2.03% |
| ECE | ↓ | 0.0138 | **0.0011** | +92.0% |

*Table 3.* Comprehensive performance comparison on the TabCredit OOT test set. AUC-ROC, Lift, and AUPRC evaluate ranking quality, while Brier Score and ECE assess probability calibration.

## 4.2. SOTA Performance on Results

**Comparison on Public Benchmarks:** We compared ArbNN with mainstream neural models, tree models and hybrid models on the public datasets. Results are presented in Table 1. As shown, ArbNN achieves SOTA results on nearly all benchmark datasets, consistently outperforming other models, including XGBoost and LightGBM. The key to this superior performance is two-fold: effective initialization and end-to-end optimization. During initialization, ArbNN inherits the predictive power of pre-trained ensemble decision trees, positioning it at least on par with the tree models. Crucially, it then benefits from global, end-to-end fine-tuning, surpassing the local and greedy optimization typical of ensemble methods. This optimization enables it to refine split thresholds, structural weights, and leaf values holistically.

**Comparison on TabCredit:** Given that TabPFNv2 (Hollmann et al., 2025) is primarily intended for inference on small to medium sized datasets, we evaluate it on TabCredit using a stratified subsample of 10,000 instances as the inference context, ensuring a proportional class distribution consistent with the full dataset. It can be seen from Table

| Trainable parameters | M2+@MOB3 | M2+@MOB6 | M2+@MOB12 | AUC-ROC | Interpretability |
|---|---|---|---|---|---|
| None (Decompiled) | 53.92 | 48.19 | 39.13 | 81.88 | Preserved |
| All parameters | 53.69 | 48.09 | 39.04 | 81.82 | Lost |
| Tree-structure matrix P | 53.94 | 48.18 | 39.12 | 81.88 | Partially lost |
| Leaf values v | 54.10 | 48.38 | 39.30 | 81.91 | Preserved |
| Bias $f$ | 54.08 | 48.37 | 39.25 | 81.90 | Preserved |
| $f$ + v + P | 54.11 | 48.40 | 39.28 | 81.93 | Partially lost |
| $f$ + v (Randomly Init) | 53.95 | 48.20 | 39.17 | 81.90 | Preserved |
| $f$ + v (Decompiled Init) | 54.13 | 48.43 | 39.32 | 81.94 | Preserved |

*Table 4.* Study on the effect of fine-tuning different parameter subsets in ArbNN on the TabCredit dataset.

2 that ArbNN consistently achieves the highest KS values across all MOB points, outperforming the suboptimal baseline by margins of around 0.5% or more at each stage. This underscores ArbNN's stronger ability to discriminate between potential defaulters and non-defaulters over time, especially in a large-scale and highly heterogeneous credit-risk environment.

While KS, AUC-ROC, and Lift are standard metrics for tracking discrimination decay in vintage analysis, evaluating highly imbalanced credit datasets necessitates cost-sensitive and calibration-aware metrics. As summarized in Table 3, we evaluate ArbNN against the strongest GBDT baseline, XGBoost, across five key dimensions on the TabCredit OOT test set. Regarding ranking quality, ArbNN consistently outperforms XGBoost in AUC-ROC (0.8194 vs. 0.8188), Lift (1.8442 vs. 1.8440), and AUPRC (0.1168 vs. 0.1166). Although these numerical gains are marginal, they indicate a robust ability to prioritize high-risk borrowers. More significantly, ArbNN demonstrates superior probability calibration, achieving a 2.03% reduction in Brier Score and a substantial 92.03% improvement in Expected Calibration Error (ECE). These results confirm that ArbNN provides not only accurate risk rankings but also highly reliable probability estimates, which are essential for real-world financial decision-making.

**Comparison on Our Private Commercial Dataset:** To rigorously evaluate OOT generalization, we conducted a vintage analysis spanning January to October 2024. Results are given in Figure 2. Despite significant temporal drift (10-month shift) and target heterogeneity (MOB2–MOB6) relative to the training set (Dec 2023), ArbNN consistently outperformed XGBoost. It achieved stable gains with an average relative improvement of $\approx 1.2\%$ and an absolute KS increase of $\approx 0.6$. These results confirm ArbNN's superior robustness and adaptability to evolving real-world data distributions compared to traditional GBDTs.

To contextualize the practical significance of these numerical gains, within our highly active premium customer segment (comprising approximately 3 million users with a credit portfolio balance in the tens of billions of dollars), a 1% relative increase in the KS statistic roughly translates to a 0.5% to 0.6% reduction in overall annualized risk. Furthermore, ArbNN is already successfully deployed in this real-time commercial trading environment. It comfortably satisfies strict production constraints, operating with under 100ms latency, a memory footprint of less than 20MB, and reliably handling normal peak loads of 20–40 QPS. These results confirm ArbNN's superiority over traditional GBDTs in both predictive robustness and system efficiency.

### 4.3. Ablation and Interpretability Analysis

**Bidirectional Fidelity and Training Dynamics:** We first verify the bidirectional fidelity of ArbNN by evaluating the *None (Decompiled)* configuration in Table 4, which tests the decompiled symbolic tree without any additional trainable parameters. Fundamentally, a traditional decision tree employs *hard routing*, deterministically assigning a sample to exactly one leaf. Conversely, ArbNN utilizes *soft routing* via softmax, where every leaf receives a continuous activation weight, although the target leaf commands an exponentially dominant share. While this architectural distinction causes their absolute output values to naturally differ, the soft neural inference and the hard symbolic inference are nearly perfectly aligned in their ranking behavior on large-scale datasets. Specifically, both KS and AUC metrics agree up to 4–5 decimal places. This rigorous alignment confirms that extracting a discrete, interpretable tree from the neural architecture preserves predictive fidelity, proving that interpretability and performance are aligned rather than traded off.

To further assess the optimization dynamics, we examined the $f + v$ *(Random Init)* setting, where split thresholds $f$ and leaf values $v$ are randomly initialized while the tree topology is kept fixed. Even under this uninformed initialization, ArbNN reliably converges and achieves performance that exceeds the XGBoost baseline. This demonstrates that on top of a fixed tree topology, the differentiable ArborCell formulation is able to optimize toward a strictly better solution,

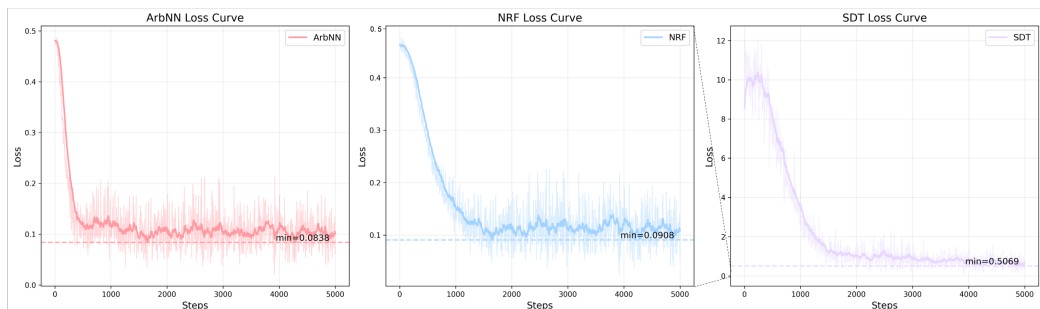

*Figure 3.* Training loss comparison under randomly initialized $f$ and $v$. ArbNN converges faster and to a lower loss than prior neuralized–tree models (NRF and SDT), illustrating the stability and efficiency of its one-shot full-path routing formulation.

revealing that our continuous relaxation not only preserves the tree structure but also yields a more expressive and trainable refinement of it. The loss-curve analysis in Figure 3 further corroborates this result: with randomly initialized $f$ and $v$, ArbNN converges faster and more stably than prior neuralized–tree models such as NRF (Biau et al., 2019) and SDT (Frosst & Hinton, 2017), highlighting the advantages of its one-shot full-path routing and well-conditioned optimization geometry.

**Parameters Fine-Tuned Study:** While all parameters within the ArborCell architecture are fully differentiable, we conducted ablation studies on the TabCredit dataset to identify the optimal fine-tuning strategy (Table 4). Our results demonstrate that fine-tuning only the split thresholds and leaf values—denoted as $f + v$ *(Decompiled Init)*—achieves the best overall performance, yielding the highest KS and AUC-ROC, while strictly preserving the original decision-path logic and full model interpretability. Conversely, unfreezing all parameters degrades performance and completely eliminates interpretability by destroying the tree-aligned sparse structure. Similarly, updating the tree-structure matrix alters subspace allocation and weakens semantic alignment, resulting in only partial interpretability. These findings underscore a critical architectural insight: the physical routing topology fundamentally dictates the model's representational boundaries. This dependency directly explains our design rationale for deliberately bootstrapping from strong GBDTs. By embedding a high-capacity structural prior into the initial topology, we provide a sufficiently expressive routing framework. This allows the network to effectively co-optimize thresholds and values via gradient descent, maximizing empirical performance while strictly maintaining interpretability.

**Axis-Aligned Decision Tree Perspective Interpretability:** To show the retained axis-aligned decision tree level interpretability of ArbNN, we first emphasized that ArborCell naturally supports a bidirectional mapping: the *Compilation* step converts a decision tree $\mathbf{T}$ into a differentiable Arbor-Cell representation, enabling **end-to-end training**, while

the *Decompilation* step maps the fine-tuned ArborCell back to a refined tree $\mathbf{T}'$, thereby ensuring **interpretability retention**. Crucially, this allows us to apply *TreeSHAP*—the tree-specific variant of SHapley Additive exPlanations (SHAP) (Lundberg et al., 2019)—which leverages the symbolic tree structure for exact and efficient feature attribution.

## 5. Conclusion and Discussion

The central contribution of ArbNN lies in conceptually decoupling tabular learning into two distinct phases: *structure learning* and *parameter learning*. In its current instantiation, ArbNN extracts the physical topology from strong ensemble decision trees (e.g., XGBoost) and maps it into a fully end-to-end differentiable neural architecture. By freezing this high-capacity structural prior, ArbNN leverages continuous gradient descent to globally co-optimize split thresholds and leaf weights. This paradigm successfully marries the robust inductive bias of tree-based models with the optimization advantages of neural networks, culminating in a framework that delivers state-of-the-art predictive performance while strictly preserving interpretability.

While relying on a pre-established topology ensures stability, a natural objective for future work is the fully differentiable, from-scratch learning of both structure and parameters. However, our preliminary explorations indicate that this is highly non-trivial. Attempting to learn the tree-structure matrix ($\mathbf{P}$) from scratch via gradient descent currently yields suboptimal results. Without the strict topological constraints imposed by the tree prior, the routing matrix rapidly degenerates into a dense, unconstrained attention mapping. This structural collapse not only induces severe over-parameterization and optimization instability on tabular datasets but also completely obliterates the model's structural interpretability. Consequently, natively learning discrete, symbolic tree structures from scratch via continuous optimization remains a profound open challenge in the field, marking the primary frontier for our subsequent research.

## Impact Statement

This paper presents a neural-symbolic learning framework aimed at improving the accuracy, stability, and interpretability of models for tabular data. While our experiments include credit risk prediction as a representative application, the proposed ArbNN is a general machine learning method rather than a decision-making system. As with other predictive models deployed in high-stakes domains, potential risks include misuse, over-reliance on automated predictions, and biases introduced by data or deployment practices. We emphasize that ArbNN is designed to improve transparency and auditability compared to black-box models, and its responsible use requires appropriate human oversight and domain-specific regulatory compliance. We do not foresee ethical concerns beyond those commonly associated with applied machine learning models.

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

## A. Tree Parsing Algorithm

To bridge gradient-boosted decision trees (GBDTs) and the proposed Arboreal Neural Network (ANN), we designed a tree parsing algorithm that systematically converts each tree in a GBDT ensemble into a differentiable ArborCell representation. The procedure ensures that the original inductive bias and structure of the tree are faithfully preserved, while making the resulting model fully trainable under end-to-end optimization. The algorithm consists of the following key steps:

---

**Algorithm 1** Tree Parsing Algorithm

---

**Input:** Decision tree $T$; input dimension $t$

**Output:** Number of inner nodes $n_i$, leaf nodes $n_j$; structure matrix $\mathbf{P}$, weight matrix $\mathbf{W}$, splits vector $f$, leaf values $\mathbf{v}$, leaf node set $\mathcal{L}$

**Initialize:** leaf values $\mathbf{v}$, leaf node set $\mathcal{L}$, internal node set $\mathcal{I}$, split features $s$, split thresholds $f$, path record $\mathcal{P}$, direction record $\mathcal{D}$; counters $l \leftarrow 0$, $i \leftarrow 0$

**Define recursive function DFS$(n, \mathcal{P}, \mathcal{D})$: begin**

  **if** *n is a leaf* **then**

    **if** $n.id \notin \mathcal{L}$ **then**

      $\mathbf{v}[n.id] \leftarrow n.\text{leaf\_value}$; $\mathcal{L}[n.id] \leftarrow l$; $l{+}{+}$

    $\mathcal{P}[n.id] \leftarrow \mathcal{P} + [n.id]$; $\mathcal{D}[n.id] \leftarrow \mathcal{D}$

  **else**

    **if** $n.id \notin \mathcal{I}$ **then**

      $\mathcal{I}[n.id] \leftarrow i$; $s[n.id] \leftarrow n.\text{feature\_index}$; $f[n.id] \leftarrow n.\text{split\_threshold}$; $i{+}{+}$

    **foreach** *child $n'$ in $n.children$* **do**

      $\mathcal{D}' \leftarrow \mathcal{D} + [\delta(n', n)]$; **DFS**$(n', \mathcal{P} + [n.id], \mathcal{D}')$

**Function** $\delta(n', n) = \begin{cases} 1 & \text{if } n'.id = n.\text{right\_child} \\ -1 & \text{otherwise} \end{cases}$

**Main Procedure: begin**

  **DFS**$(T, [\,], [\,])$; $n_i \leftarrow |\mathcal{I}|$, $n_j \leftarrow |\mathcal{L}|$

  Initialize $\mathbf{W} \in \mathbb{R}^{n_i \times t}$, $f \in \mathbb{R}^{n_i}$ with zeros

  **foreach** $(n.id, idx) \in \mathcal{I}$ **do**

    $\mathbf{W}_{idx, s[n.id]} \leftarrow 1$; $f_{idx} \leftarrow c[n.id]$

  Initialize $\mathbf{P} \in \mathbb{R}^{n_j \times n_i}$ with zeros

  **foreach** $(n.id, l') \in \mathcal{L}$ **do**

    $p \leftarrow \mathcal{P}[n.id]$; $d \leftarrow \mathcal{D}[n.id]$

    **foreach** $(nid, d_i) \in (p, d)$ **do**

      **if** $nid \in \mathcal{I}$ **then**

        $\mathbf{P}_{l', \mathcal{I}[nid]} \leftarrow d_i$

  $\mathbf{P} \leftarrow$ **ApplyWeights**$(\mathbf{P})$; reorder $\mathbf{v}$ based on $\mathcal{L}$

  **return** $n_i, n_j, \mathbf{P}, \mathbf{W}, f, \mathbf{v}, \mathcal{L}$

**Function ApplyWeights$(\mathbf{P})$: begin**

  $k \leftarrow$ max number of non-zeros per row in $\mathbf{P}$; $w_j \leftarrow (1/2)^j$ for $j = 0, \dots, k-1$

  **foreach** *row $i$ in $\mathbf{P}$* **do**

    $I_{nz} \leftarrow$ indices of non-zero elements in $\mathbf{P}_i$

    **foreach** $(j, idx) \in$ *enumerate*$(I_{nz})$ **do**

      $\mathbf{P}_{i, idx} \leftarrow \mathbf{P}_{i, idx} \cdot w_j$

  **return** $\mathbf{P}$

---

## B. TabCredit Dataset

To overcome the limitations of existing small-scale academic benchmarks, we introduce **TabCredit**, a proprietary commercial credit dataset covering a two-month period, comprising approximately **1 million unique user loan application records per month**, totaling around **2 million records**. Each record has 100 anonymized and extensively engineered features.

TabCredit is partitioned into two parts: a *training set* consisting of users with successful loan applications from September 2023, and an *out-of-time (OOT) test set* containing users from November 2023. This temporal split enables strict in-time and OOT evaluation, ensuring rigorous assessment of model generalization. For each record, TabCredit provides future risk outcomes for both training and OOT sets, including monthly delinquency indicators from two to twelve months-on-book (MOB2–MOB12). Here, **MOB** refers to the number of months after loan origination, and delinquency is measured using the industry-standard **M2+** definition, which denotes accounts with payments overdue by *31 days or more*. Accordingly, the reported rates correspond to the cumulative proportion of users who have ever entered the M2+ state by each MOB horizon.

Table 5 illustrates representative cumulative M2+ delinquency rates across two cohorts. As expected, delinquency starts at relatively low levels (around 0.2–0.3% at MOB2), but grows monotonically with MOB, reaching approximately 5–6% by MOB12. This steady increase reflects the temporal dynamics of credit portfolios and provides a realistic setting for developing and evaluating risk forecasting models.

| Cohort | MOB2 | MOB3 | MOB4 | MOB5 | MOB6 | MOB7 | MOB8 | MOB9 | MOB10 | MOB11 | MOB12 |
|---|---|---|---|---|---|---|---|---|---|---|---|
| 202309 | 0.26% | 0.67% | 1.35% | 2.05% | 2.66% | 3.40% | 4.04% | 4.62% | 5.16% | 5.64% | 6.16% |
| 202311 | 0.20% | 0.67% | 1.26% | 1.82% | 2.57% | 3.22% | 3.80% | 4.33% | 4.86% | 5.40% | 5.92% |

*Table 5.* Example cumulative **M2+** delinquency rates (defined as 31+ days past due) across different months-on-book (MOB2–MOB12) for two representative cohorts in TabCredit. Delinquency accumulates steadily with MOB, highlighting the temporal progression of credit risk.

In addition to overall dataset statistics, we provide representative examples of feature characteristics from TabCredit in Table 6. The dataset contains heterogeneous attributes drawn from both credit bureau records and behavioral signals. For instance, **Credit Bureau Main Cardholder Score** (99% coverage) captures a user's repayment history, while **Credit Bureau Loan Record Last Update Date** (99%) reflects the freshness of a borrower's bureau file. Utilization-related indicators, such as the **Max Credit Utilization Ratio (Past 3 Months, 100% coverage)** and the **Current Fixed Credit Limit (100%)**, characterize users' recent borrowing behavior and credit capacity. Finally, the **Number of Inquiry Institutions in the Past 6 Months** (coverage $\sim$87%) measures the intensity of recent credit-seeking behavior, often serving as an early warning indicator of potential default risk. Together, these features illustrate the diversity, realism, and predictive richness of TabCredit.

| Feature | Type | Coverage | Range / Example Values | Description |
|---|---|---|---|---|
| Credit Bureau Main Cardholder Score | Numerical (ordinal) | 99% | 300–900 (e.g., 650, 720) | Bureau-based credit score for existing cardholders |
| Credit Bureau Loan Record Last Update Date | Date / Timestamp | 99% | e.g., 2023-07-15 | Last update date of loan record in credit bureau |
| Max Credit Utilization Ratio (Past 3 Months) | Numerical (ratio) | 100% | 0–1 (e.g., 0.25, 0.83) | Maximum utilization ratio of credit line in past 3 months |
| Current Fixed Credit Limit | Numerical (continuous) | 100% | 0–300k (e.g., 50,000; 120,000) | Current fixed credit line granted to the user |
| Number of Inquiry Institutions in the Past 6 Months | Numerical (count) | 87% | 0–15 (e.g., 2, 6) | Count of distinct institutions inquiring about user's credit in the past 6 months |

*Table 6.* Representative feature characteristics from the TabCredit dataset. Coverage denotes the proportion of users with non-missing values for each feature. Notably, frequent inquiries (last row) often serve as strong risk indicators in real-world credit scoring.

In summary, TabCredit combines scale, feature richness, temporal partitioning, and longitudinal outcomes. It provides a realistic and challenging benchmark for evaluating not only predictive accuracy, but also interpretability and stability of tabular learning methods in credit risk modeling.

