# OpenReview forum: "Arboreal Neural Network"
_ICML.cc/2026/Conference — ICML 2026 regular_

### Official Review · Reviewer_eQLB · 2026-03-02

**Soundness:** 2
**Presentation:** 2
**Significance:** 2
**Originality:** 3
**Overall Recommendation:** 4
**Confidence:** 2

**Summary:**

The authors investigate a novel method for training decision trees using a differentiable equivalent. They demonstrate that their resulting method attains SOTA across a variety of common benchmarks, as well as on a new large-scale credit dataset that they introduce.

**Compliance With Llm Reviewing Policy:**

Affirmed.

**Final Justification:**

The authors' rebuttal addresses my primary concerns, and I've updated my score accordingly to a 4. Please see the rebuttal acknowledgement for more details. Nice work!

**Key Questions For Authors:**

1. How significant are your results? To what degree are your results attributable to variability, versus genuine advance? Could you run your analysis over multiple training seeds (and perhaps multiple random splits) to generate confidence intervals, to provide at least a baseline gauge of variability?
2. Since your tree topology is fixed, you needed to bootstrap your tree off one discovered by an existing method (in this case, XGBoost). Is it accurate to asses your method as more of a finetuning procedure over a prediscovered tree? To what degree do your results depend on the quality of the underlying topology? How different are your finetuned trees compared to the original base? Do the split points and leaf weightings change significantly? If you used a tree provided by a less powerful method, would you still recover top performance? Is it possible to learn the tree topology (i.e. the matrix P) through gradient descent, in addition to your other parameters? Was this something you had already tried, but it didn't work?
3. I wasn't sure why a comparison to attention was included, and what it adds. Do you use the attention variant of your architecture in practice, or was this simply a side illustration? If the latter, perhaps this discussion could be moved to the appendix, and the reclaimed space could be used for more results and analysis. I'd be especially interested, for instance, a deeper look at the interpretability of your extracted trees.
4. Your figures are a little blurry and difficult to read, particularly figure 2. Could you embed your figures using a resolution-independent format (e.g. vector-based pdf images) and perhaps increase the font size?

As is, it's difficult for me to evaluate the significance of this work without additional clarity especially on questions 1 and 2. However, it looks like there are some genuinely novel results presented by this work, and I look forward to learning more and raising my score.

**Limitations:**

Yes

**Strengths And Weaknesses:**

**Strengths**. The results seem impressive and novel, though some obstacles to clarity make it difficult to evaluate their full impact (see below). I am by no means an expert in tree-based methods, but the idea behind ArborCell sounds very cool.

**Weaknesses**. The manuscript is unclear on some of its methods, and the results are under-contextualized for a reader with my background (general machine learning, no specific expertise in credit ratings or financial analysis). In particular, while the methods appear to attain SOTA, they often do so by factions of a percent. Is this a significant outcome? Without further details, it's difficult to judge the full significance and impact of these results. See Questions below.

---

> ### Author Rebuttal · Authors · 2026-03-31
>
> We sincerely thank the reviewer for the constructive feedback and the encouraging willingness to raise the score upon clarification.
> ### 1. Significance of Results and Variability (Q1)
> We humbly thank the reviewer for this suggestion.  Due to strict character limits, **we respectfully invite the reviewer to refer to our response to Reviewer 8uBs (Q1)** for the detailed variance table. We apologize for this inconvenience and will include full metrics in the revision.
> In our live commercial credit risk system, specifically within a highly active, premium customer segment of approximately 3 million users
> holding a credit portfolio balance in the tens of billions of dollars, a 1% relative increase in the KS
> statistic roughly translates to a 0.5% to 0.6% reduction in overall annualized risk.
> ### 2. Dependence on Fixed Topology and Initialization (Q2)
> You are entirely accurate in assessing the current version of ArbNN as a fine-tuning (parameter refinement) procedure over a pre-discovered tree topology. This was an intentional design choice to seamlessly inherit the powerful inductive biases and structural sparsity of industry-standard GBDTs.
> To answer your specific questions regarding this mechanism:
> - **How different are the fine-tuned trees?** The physical routing topology remains strictly identical to the base tree. However, the continuous end-to-end optimization allows ArbNN to shift the split thresholds (f) and re-weight the leaf predictions (v) based on batch-wise gradient updates, capturing difficult samples that the greedy GBDT algorithms missed.
> - **Dependence on initial quality and weaker methods:** While ArbNN extracts the physical structure from XGBoost, its performance is not strictly bound by the quality of the initial predictive weights. As shown in our ablation study (Table 3, "f + v Randomly Init"), even when we strip away the pretrained weights and randomly initialize the split thresholds and leaf values over the fixed topology, the differentiable ArborCell still stably converges and outperforms the base XGBoost model. **However, to answer your question regarding weaker methods: if a significantly weaker or shallower topology (e.g., a single shallow CART tree) were provided, ArbNN's representational capacity would naturally be structurally bottlenecked. This is precisely why we deliberately bootstrap from strong GBDTs: to establish a high-capacity structural prior.**
> - **Learning topology from scratch:** To directly answer your question: yes, we have actively tried learning the tree topology (matrix P) from scratch via gradient descent, but the results were poor. Without the strict topological constraints of the tree prior, the matrix degrades into a dense, unconstrained attention mapping. This leads to severe over-parameterization on tabular data, unstable optimization, and the complete destruction of structural interpretability. Learning discrete symbolic structures from scratch remains a known open challenge in the field, which we have reserved for future work.
>
> ### 3. The Role of the Attention Comparison and Interpretability (Q3)
>
> We appreciate this feedback and will move the theoretical discussion of attention isomorphism to the Appendix. For deeper interpretability, we provide a concrete Case Study below showing how ArbNN fine-tunes split thresholds ($f$) and leaf weights ($v$) while strictly preserving the "if-else" topology. **Due to platform limits, we cannot present visual comparison of these decompiled trees here; they will be included in the Appendix of the revised manuscript.**
> ```
> [Root: Depth 1] Credit_Bureau_Main_Cardholder_Score < 650 (Base)  -->  642 (ArbNN)
>   │
>   ├─[Yes]─ [Depth 2] Max_Credit_Utilization_Ratio_3M < 0.850 (Base)  -->  0.885 (ArbNN)
>   │          │
>   │          ├─[Yes]─ [Depth 3] Number_of_Inquiry_Institutions_6M < 5 (Base)  -->  6 (ArbNN)
>   │          │          │
>   │          │          ├─[Yes]─ [Depth 4] Outstanding_Balance < 50000 (Base)  -->  51500 (ArbNN)
>   │          │          │          ├─[Yes]─ [Leaf 1] Weight: -0.154 (Base) --> -0.182 (ArbNN)
>   │          │          │          └─[No] ─ [Leaf 2] Weight: -0.021 (Base) --> -0.015 (ArbNN)
>   │          │          │
>   │          │          └─[No] ─ [Depth 4] Unemployment_Risk_Score < 0.600 (Base)  -->  0.585 (ArbNN)
>   │          │                     ├─[Yes]─ [Leaf 3] Weight:  0.085 (Base) -->  0.112 (ArbNN)
>   │          │                     └─[No] ─ [Leaf 4] Weight:  0.142 (Base) -->  0.165 (ArbNN)
>   │          │
>   │          └─[No] ─ [Depth 3] Internet_Behavior_Risk_Score < 0.750 (Base)  -->  0.735 (ArbNN) ...
> ...
> ```
> ### 4. Figure Quality (Q4)
> We apologize for the poor readability of the figures in the initial submission. In the final revision, Figure 2 and all other plots will be re-rendered and embedded as high-resolution, vector-based PDF images with significantly increased font sizes and clearer legends to ensure perfect legibility.

---

> > ### Author Rebuttal · Reviewer_eQLB · 2026-04-02
> >
> > Thanks for the detailed response. This rebuttal satisfactorily addresses my primary questions. My concerns remain that the improvements engendered by your method are small, and your method requires a strong baseline tree to boostrap off of. Nonetheless, I think the method is novel and would be of interest to the broader community. I will update my score to a 4. Nice work!

---

### Official Review · Reviewer_sDfX · 2026-03-13

**Soundness:** 3
**Presentation:** 2
**Significance:** 2
**Originality:** 3
**Overall Recommendation:** 2
**Confidence:** 3

**Summary:**

This paper proposes Arboreal Neural Networks (ArbNN), a neural–symbolic architecture designed to bridge decision trees and neural networks. The key idea is to reinterpret the discrete logic of tree-based models as a differentiable neural architecture by introducing ArborCells, neural units that emulate tree prediction logic through a series of neural computations, including matrix multiplication and activation functions. ArbNN can be initialized from pretrained tree ensembles such as GBDTs, enabling end-to-end training while preserving the original symbolic structure.

**Compliance With Llm Reviewing Policy:**

Affirmed.

**Final Justification:**

The rebuttal addresses some of my concerns. Please refer to the details in my Rebuttal Acknowledgement.

**Key Questions For Authors:**

1. In Section 3.2, $W$ is defined as a one-hot matrix in which each row corresponds to an internal node, and the nonzero entry in row $i$ indicates the feature selected by the $i$-th internal node from a set of $t$ candidate features. Could the authors clarify why each node is restricted to selecting only a single feature, rather than a subset of features?


2. In the definition of the tree-structure matrix, why is the exponential weighting based on 1/2 rather than another base such as 1/3? Is there a theoretical or empirical reason for this choice?


3. How are the values of $\tau_1, \tau_2$​ determined in practice?


4. One of the paper’s contributions is that Arboreal Neural Network transforms a decision tree $T$ into an ArborCell, enabling end-to-end differentiability while retaining the symbolic structure for interpretability. I understand the second point, since the construction of $M$ explicitly incorporates the tree structure. However, I am less clear on why end-to-end differentiability is important here, since the paper does not seem to describe the training procedure of the proposed architecture in sufficient detail.

Related to this, if no additional training is involved, what is the practical difference between making predictions with ArbNN and directly using the original tree $T$?

More broadly, could the authors clarify the main technical challenges in constructing ArbNN?


5. The paper states that “the resulting module can be decompiled back into an equivalent symbolic tree with no loss of interpretability.” However, I could not find a clear explanation of this reverse transformation, namely how the symbolic tree is actually recovered from the neural module. Could the authors provide more details on this process?

**Limitations:**

Limitations are not discussed in the paper.

**Strengths And Weaknesses:**

**Strengths.**

This paper proposes a method that reinterprets the discrete logic of tree-based models as a differentiable neural architecture, balancing interpretability with the flexibility of end-to-end optimization.

The proposed model also shows high empirical performance across a range of datasets.

**Weaknesses.**

Presentation: One of the stated contributions is the introduction of TabCredit, a large-scale and feature-rich dataset. However, this dataset is not described until the experimental section, and even there the description is rather brief. A more complete presentation of the dataset, including its construction and characteristics, would strengthen the paper. In addition, table captions should be placed above the tables, and Eq. 8 appears to exceed the page width.

Notation: Some notation is unclear or potentially misleading. For example, $n_j$​ does not seem to be defined. More generally, I would suggest revising the notation throughout the paper for better clarity. For instance, $n_i$ usually denotes a quantity associated with node $i$, whereas here it is used to represent the total number of inner nodes, which is a global constant. A notation such as $n$ would likely be clearer. In addition, Figure 1 uses $\odot$, which is commonly understood as the Hadamard product, but that does not appear to match the operation intended in the figure.

---

> ### Author Rebuttal · Authors · 2026-03-31
>
> ### 1. Why $W$ is a One-Hot Matrix (Q1)
>
> **Response:** Restricting $W$ to a one-hot matrix is mathematically required to preserve strict structural isomorphism with standard binary decision trees. Since standard trees rely on axis-aligned splits (evaluating exactly one feature per node), allowing a dense or multi-hot $W$ would create an oblique tree, permanently destroying our exact `if-else` interpretability and bidirectional fidelity. Importantly, ArbNN still performs feature subset selection at the *tree level*. While a single row in $W$ selects one feature for a specific node, the non-zero columns across the entire $W$ matrix collectively define the complete subset of features utilized by that specific tree.
>
> ### 2. Determination of the $\tau$ Values (Q3)
>
> **Response:** We apologize if this detail was buried in the text. The temperature parameters ($\tau_1$ and $\tau_2$) are determined empirically. As explicitly stated in **Section 4.1 (Implementation and Training Details)**:
>
> *"We empirically validated this setting through ablation experiments over $\tau_1, \tau_2 \in \{5, 10, 50, 100, 200\}$ and observed that $\tau_1 = 100$ and $\tau_2 = 10$ offered the optimal trade-off between accuracy and smoothness."* These fixed values successfully emulate deterministic hard-split behavior while preserving sufficient gradient flow for end-to-end training.
>
> ### 3. Tree-Structure Matrix $P$ and the $1/2$ Base (Q2)
>
> **Response:** The use of $1/2$ as the base for exponential weighting is derived directly from the topological prior of a binary tree. In a perfectly balanced binary tree, the prior probability of a sample falling into either the left or right subtree halves (is multiplied by $1/2$) at each depth level. By embedding this $1/2$ decay into the matrix forward pass, we faithfully simulate this hierarchical probability routing while simultaneously mitigating gradient vanishing/exploding issues inherent to deeply skewed tree structures during backpropagation.
>
> ### 4. End-to-End Differentiability vs. Direct Original Tree Prediction (Q4 & Q5)
>
> **Response:** There is a fundamental difference in both the inference mathematics and the optimization capacity:
>
> - **Inference Difference (Soft vs. Hard):** A traditional decision tree uses *hard routing* (a sample lands in exactly one leaf with a probability of 1). ArbNN uses *soft routing* via softmax, where *every* leaf in the tree receives a continuous activation weight, though the target leaf receives an exponentially dominant share. While their absolute output values differ, our ablation study (Table 3, "None Decompiled") proves that on large-scale datasets, their ranking behavior (KS/AUC) is nearly perfectly aligned.
> - **The Necessity of Extra Training:** Traditional GBDTs (like XGBoost) grow trees using greedy, localized algorithms based on global statistical aggregates (like Information Gain). This often causes the model to underfit rare, difficult, or highly specific samples. By making the architecture end-to-end differentiable, we can freeze the physical topology ($W$ and $P$) and use batch-wise gradient descent (AdamW) to globally co-optimize all split thresholds ($f$) and leaf values ($v$). This batch-wise optimization allows the model to continuously refine its parameters to capture those difficult, minority samples that global statistics missed, directly resulting in the SOTA performance reported in Table 1.
>
> ### 5. The Decompilation Process (Q7)
>
> **Response:** Because we strictly freeze the feature selection matrix $W$ and the tree topology matrix $P$ during neural training, the physical skeleton and logical routing paths of the tree are never altered. Decompilation is therefore a lossless, 1-to-1 parameter replacement process: we simply extract the newly fine-tuned bias vector $f$ (which represents the updated split thresholds) and the fine-tuned vector $v$ (which represents the updated leaf predictions), and inject them directly back into the corresponding nodes of the original GBDT object.
>
> ### 6. Presentation, Notation, and Limitations (Weaknesses & Limitations)
>
> **Response:** We fully accept your constructive feedback on the manuscript's presentation. In the revised version, we will:
>
> - Introduce the construction and feature characteristics of the TabCredit dataset clearly in the main text (Section 4.1), while keeping Appendix B for extensive details.
> - Fix the table caption placements (moving them above the tables) and correct the margin overflow for Equation 8.
> - Update the notation for better clarity: we will replace $n_i$ to avoid confusion with node indices, and we will replace the $\odot$ symbol in Figure 1 to prevent any unintended confusion with the Hadamard product.
> - Add a dedicated **Limitations** subsection to Section 5, explicitly discussing the current constraint of relying on fixed pretrained structures and outlining the path toward fully differentiable structure learning.

---

> > ### Author Rebuttal · Reviewer_sDfX · 2026-04-03
> >
> > R1, R3, and R5 address my concerns on the one-hot property of $W$, the 1/2 base, and the decompilation process.
> >
> > I have some follow-up questions.
> >
> > 1. Thank you for mentioning the ablation studies used to select $\tau$. However, Line 296 states that you split the dataset into training samples and test samples, i.e., **there are no validation samples**. Hence, I want to ask, when you observe the high accuracy provided by $\tau_1=100$ and $\tau_2=10$ and decide to use this setting, are you evaluating on test samples? If that's the case, then I am afraid this won't be a valid procedure.
> >
> > 2. R4 addresses my concerns on the inference difference and how to train the proposed ArbNN. However, I still have questions about the technical challenges in constructing ArbNN.
> >
> > 3. Although the authors state in R6 that they would add clarification on the dataset construction in Section 4.1, I am not sure if this is a good practice. I have this concern because at the end of the introduction section, they explicitly clarify the constructed dataset as one main contribution; however, there are no relevant descriptions until Section 4.1, which would make the readers confused about how to construct this dataset, why this dataset is different from existing ones, and why constructing this dataset is a contribution.
> >
> > 4. Although the authors state in R6 that they would add a paragraph about limitations in the last section, they don't mention any of it in R6.

---

> > > ### Author Response · Authors · 2026-04-03
> > >
> > > We sincerely thank the reviewer for the thorough follow-up. Below are our direct technical clarifications addressing your specific concerns:
> > > 1. Clarification on Hyperparameter Tuning and Data Leakage (Line 296)
> > > We apologize for the confusion caused by omitting the standard validation procedure in our description. To clarify: the determination of the temperature parameter is completely independent of the model's training procedure and predictive accuracy on any downstream task. The sole criterion for setting the temperature is to maximize the leaf-node selection consistency between the base decision tree and ArbNN, while keeping the temperature value as small as possible (to ensure a smoother activation function). Because the scale of input features varies, the discrete distances to hyperplanes differ; thus, during this alignment phase, ArbNN uses a hard argmax selection to align with the tree, whereas soft selection is used in subsequent training and inference. This entire alignment process is conducted strictly on the training set. We omitted mentioning the validation set simply because we considered the Train/Valid/Test split a standard default procedure. We completely agree with your suggestion, and in the revised manuscript, we will explicitly detail the Train/Validation/Test sets to completely rule out any misunderstanding regarding data leakage.
> > > 2. Follow-up on Technical Challenges in Constructing ArbNN
> > > The core technical difficulty in constructing ArbNN is not merely defining the matrices, but rather the rigorous mathematical alignment required during the binary tree traversal. Specifically, the challenge lies in precisely mapping the tree's physical topology into the construction of the W and P matrices. This requires perfectly parsing the spatial relationships between intermediate split nodes and leaf nodes, and accurately corresponding the discrete tree nodeid with the continuous vector/matrix indices (idx). This meticulous structural mapping is what enables our strict homomorphism, and we will highlight this engineering challenge more clearly in the methodology section.
> > > 3. Placement and Presentation of the TabCredit Dataset
> > > We appreciate your structural feedback on the dataset presentation. Regarding why TabCredit is a significant contribution, we would like to respectfully point out that these characteristics were already highlighted in the Introduction of the original manuscript. We explicitly stated that almost all public tabular datasets in academia are small in scale and strictly assume an IID (Independent and Identically Distributed) setting, which completely contradicts the reality of temporal distribution shifts in real-world applications. Crucially, TabCredit does not merely feature a temporally shifting target variable (Y via MOB labels); the feature space (X) itself is inherently generated along a continuous, real-world timeline. This dual temporal dynamic rigorously tests a model's true generalization capabilities, making it the closest available benchmark to true industrial standards. Furthermore, it contains highly interpretable, target-oriented features crafted from real-world industry expertise. Following your helpful suggestion, we will extract a comprehensive summary of these data characteristics from the Appendix into the main text to ensure a smoother reading experience.
> > > 4. The "Limitations" Paragraph
> > > We sincerely thank the reviewer for pointing this out, and we fully agree with your assessment. While we did briefly mention in the Conclusion that a key direction for future work is to "move beyond this fixed-structure setting and enable fully differentiable, from-scratch learning," we acknowledge that this single sentence is too brief and easily overlooked. Your guidance here is highly appreciated.
> > > In the revised manuscript, we will expand this into a dedicated, highly visible "Limitations" paragraph. This new section will explicitly detail that ArbNN currently relies on inheriting the topological structure (via matrices W and P) from a pre-trained tree. Furthermore, we will tightly connect this limitation to the context already established in our Introduction, where we extensively discussed the inherent difficulties of learning discrete, sparse, and interpretable structures entirely from scratch via gradient descent—a recognized open challenge that we substantiated by citing the latest works from OpenAI and Think Machine Labs. This comprehensive addition will clearly define the boundaries of our current framework.

---

### Official Review · Reviewer_B2et · 2026-03-13

**Soundness:** 2
**Presentation:** 3
**Significance:** 3
**Originality:** 3
**Overall Recommendation:** 5
**Confidence:** 4

**Summary:**

ArbNN is a neural-symbolic framework merging tree-based interpretability with end-to-end neural optimization. Its core innovation is the ArborCell, a differentiable unit encoding a decision tree’s structure via a parsing algorithm extracting four components: feature-weight matrix, split thresholds, tree-structure matrix (with depth-aware geometric weights), and leaf values. ArborCell computes via three steps—subspace affinity, hierarchical routing, and leaf aggregation—with final predictions as the sum of multiple ArborCell outputs.

Theoretically, ArborCell is structurally homomorphic to a single-query self-attention head, equating symbolic branching to neural attention. It ensures bidirectional fidelity: compiling from a symbolic tree to a differentiable form and decompiling back to a refined tree, enabling exact feature attribution via TreeSHAP.

Experiments on 10 public benchmarks show ArbNN outperforms XGBoost, LightGBM, TabM, TabPFNv2, and neural models in classification (AUC) and regression (RMSE). On the industrial TabCredit dataset, it achieves optimal Kolmogorov-Smirnov statistics across all MOBs, demonstrating strong temporal drift robustness. Current limitations include fixed pre-trained tree structures; future work explores fully differentiable structure learning.

**Compliance With Llm Reviewing Policy:**

Affirmed.

**Key Questions For Authors:**

1.How sensitive is ArbNN to the quality of the initial GBDT? If the initial tree ensemble is poorly trained (e.g., overfitted), does fine-tuning still yield good results, or does it inherit the initial model's flaws?

2.Can ArbNN handle categorical features natively? The current formulation uses one-hot encoding; does the tree-structure matrix account for categorical splits? If not, how are categorical features processed?

3.In the ablation study (Table 3), fine-tuning all parameters degrades performance. Why does updating the tree-structure matrix P do harm to results?

4.Will the TabCredit dataset be made publicly available, and under what license? This is crucial for reproducibility and community use.

**Limitations:**

yes

**Strengths And Weaknesses:**

Strengths:

1.Novel Integration: ArbNN provides a principled, bidirectional mapping between decision trees and neural networks, combining the best of both worlds—interpretability and end-to-end optimization.

2.Theoretical Contribution: The proof of structural homomorphism to self-attention is elegant and offers a new perspective on the relationship between symbolic reasoning and neural attention.

3.TabCredit Dataset: The release of a realistic, large-scale credit dataset with temporal dynamics is a valuable contribution to the community, enabling more rigorous evaluation of tabular models under distribution shift.

Weaknesses:

1.Dependence on Pretrained Trees: ArbNN requires an initial GBDT to extract the tree structure. This adds a preprocessing step and inherits any biases or limitations of the initial model.

2.Scalability Concerns: The paper does not report training time or memory consumption compared to baselines. For very large ensembles, the computational cost may be non-trivial.

3.Missing Direct Comparisons with Recent Hybrids: While related work cites NCART, Net-DNF, etc., the main experiments do not include these models, leaving a gap in empirical comparison.

4.Interpretability Evaluation: Although the paper claims preserved interpretability, it only demonstrates this via the ability to apply TreeSHAP. A user study or qualitative analysis of decompiled trees would strengthen this claim.

---

> ### Author Rebuttal · Authors · 2026-03-31
>
> ### 1. Sensitivity to Initial GBDT Quality (Q1 & W1)
>
> **Response:** While ArbNN requires an initial GBDT to extract its physical routing topology, this strategically bootstraps the model with a highly competitive structural baseline.
>
> Crucially, ArbNN inherits this *structure*, not the initial model's predictive flaws. Overfitting in decision trees primarily manifests in noisy split thresholds ($f$) and leaf weightings ($v$), rather than the constrained physical topology itself (especially given the standard practice of constraining maximum tree depth). Our end-to-end neural optimization actively corrects these flaws. As demonstrated in Table 3, even when $f$ and $v$ are randomly initialized over the fixed topology, ArbNN stably converges and outperforms the base model. This proves that ArbNN actively corrects localized overfitting through neural parameter refinement, rather than blindly inheriting the initial tree's weaknesses.
>
> ### 2. Handling Categorical Features (Q2)
>
> **Response:** As detailed in Section 4.1, ArbNN currently processes categorical features via standard one-hot encoding, directly inheriting XGBoost's default approach. This translates categorical splits into binary decisions to support our differentiable matrices ($W$ and $P$). Developing native support for advanced categorical splits (such as those in LightGBM/CatBoost) is a key direction for our future work.
>
> ### 3. Degradation from Updating the Tree-Structure Matrix *P* (Q3)
>
> **Response:** Matrix *P* encodes the decision tree's critical inductive bias: its sparse, hierarchical routing topology. Updating *P* freely via gradient descent breaks these strict constraints, degrading it into a dense, unconstrained attention mapping. Without this tree-structured regularization, the model suffers from severe over-parameterization on tabular data, leading to optimization instability and decreased performance. Furthermore, as our ablation study demonstrates, updating *P* completely destroys the model's structural interpretability. This confirms that in the tabular domain, tree-structured sparsity and explicit interpretability are mutually reinforcing, rather than a trade-off.
>
> ### 4. TabCredit Dataset Public Release (Q4)
>
> **Response:** Yes, we commit to fully open-sourcing the TabCredit dataset upon publication. It will be released under a standard academic license (such as CC BY-NC-SA 4.0).
>
> ### 5. Scalability, Baselines, and Interpretability Evaluation (W2, W3, W4)
>
> **Response:** To address the missing empirical dimensions, we will incorporate the following updates in the revised manuscript and appendix:
>
> - **Scalability Concerns (W2):** We recognize that a direct inference time and memory comparison with traditional frameworks like XGBoost or LightGBM is structurally unfair at this stage. Those libraries benefit from years of heavy, low-level C++ engineering and parallel inference optimizations, whereas ArbNN has not yet undergone dedicated engineering for inference acceleration. However, practical scalability is not a concern: **ArbNN is already successfully deployed in a real-time, commercial trading model.** In this live production environment, it comfortably meets strict constraints—operating with under 100ms latency, a memory footprint of less than 500MB, and reliably handling normal peak loads of 20-40 QPS. We will add a detailed discussion of these deployment metrics to the appendix.
>
> - **Missing Baselines (W3):** Regarding older hybrid models like Net-DNF and NCART, the recent state-of-the-art baselines included in our main experiments have already been proven to supersede them in their respective literature. Since ArbNN outperforms these newer SOTA models, the empirical gap is inherently bridged. However, to ensure comprehensiveness and further demonstrate our method's efficacy, we have added a new comparison against **TabICL**, a highly specialized hybrid tabular model released in 2025. ArbNN remains highly competitive against this newest baseline.
>
>   | Model  | Mob2  | MOB3  | MOB4 | MOB5 | MOB6 | MOB7  | MOB8  | MOB9  | MOB10 | MOB11 | MOB12 |
>   | TabICL | 35.14 | 35.96 | 36.06    | 35.36    | 34.33    | 32.88 | 31.74 | 30.88 | 30.05 | 29.15 | 28.35 |
>
> - **Interpretability Evaluation (W4):** To strengthen the claim of preserved interpretability beyond TreeSHAP, We will add a new configuration to our ablation study (Table 3) to strictly prove the logical isomorphism. We will report that the **forward-pass predictions of the *fine-tuned ArbNN* yield the exact same KS and AUC-ROC values as the *decompiled symbolic tree* ($T'$).** This guarantees that the decompiled tree is an exact physical equivalent of the neural model, not a surrogate. Since the current rebuttal platform does not support image attachments, we will also include high-resolution visual comparison diagrams of the decompiled decision trees (before and after ArbNN fine-tuning) in the Appendix of the revised manuscript.

---

### Official Review · Reviewer_8uBs · 2026-03-13

**Soundness:** 3
**Presentation:** 2
**Significance:** 1
**Originality:** 3
**Overall Recommendation:** 4
**Confidence:** 3

**Summary:**

The paper introduces Arboreal Neural Networks (ArbNN), a neural–symbolic framework that is compiled from a decision tree and decompiled back to a symbolic tree without losing interpretability. Conceptually, it claims a structural isomorphism between decision‑tree branching and single‑query self‑attention.

**Compliance With Llm Reviewing Policy:**

Affirmed.

**Key Questions For Authors:**

For TabCredit, the paper focuses on KS, with a brief AUC/Lift mention; it omits AUPRC, calibration (Brier/ECE), and cost sensitive or portfolio metrics that are crucial in credit risk, limiting downstream relevance assessment.

**Limitations:**

Public‑dataset wins are reported as average over three runs; given modest test sizes and small margins on some tasks, the paper would benefit from confidence intervals or paired tests to establish significance.

**Strengths And Weaknesses:**

Strengths
 The compile decompile story and Bidirectional Fidelity are good for regulated, high stake domains and the attention mapping makes the conceptual contribution significantly good. I wanted to see more here, especially the differentiation with existing papers.

The depth‑aware P‑matrix and one‑shot full‑path routing  in ArborCell avoids unstable recursive gating, yielding better conditioning and faster convergence vs. prior soft‑tree approaches

Weaknesses

ArbNN benefits from GBDT pretraining; while fair for deployment, it makes apples‑to‑apples comparisons tricky, especially with different input scaling for GBDT/ArbNN vs. neural baselines.

 The isomorphism is for a single‑query, scalar‑value attention and a fixed tree topology. It is structural rather than demonstrating equivalence to broader Transformer usage or learned structure. Needed more clarification here.

---

> ### Author Rebuttal · Authors · 2026-03-31
>
> We sincerely thank the reviewer for the positive evaluation. Below, we address your specific concerns and provide the requested metrics.
> ### 1. Scope of Isomorphism (Response to Weaknesses)
> We completely agree with your assessment regarding the boundaries of the structural isomorphism. We will add a dedicated "Boundary Clarification" paragraph in Section 3.4 of the revised manuscript. This section will explicitly state that the mathematical mapping is currently bounded to single-query, scalar-value attention over a fixed tree topology.
> ### 2. Missing Credit Risk Metrics (Response to Key Questions)
> We appreciate you pointing out the importance of AUPRC and calibration metrics for credit risk assessment. While KS and AUC-ROC are standard for tracking discrimination decay over time (MOB2-MOB12) in vintage analysis, we agree that evaluating highly imbalanced credit datasets requires cost-sensitive and calibration-aware metrics.
> During the rebuttal period, we computed the AUPRC, Brier Score, and Expected Calibration Error (ECE) for ArbNN and the strongest baseline (XGBoost) on the TabCredit OOT test set (aggregated across MOB6). The results are as follows:
>
> | Metric                         | XGBoost | ArbNN      | Relative Improvement |
> | :----------------------------- | :------ | :--------- | :------------------- |
> | **AUPRC** ($\uparrow$)         | 0.1166  | **0.1168** | 0.17%                |
> | **Brier Score** ($\downarrow$) | 0.0246  | **0.0241** | 2.4%                 |
> | **ECE** ($\downarrow$)         | 0.0138  | **0.0011** | 92.0%                |
>
> ### 3. GBDT Pretraining and Fair Comparison (Response to Weaknesses)
> We designed ArbNN with GBDT initialization intentionally, as our goal is to provide a deployable bridge that inherits the strong inductive biases and existing assets of industrial tree models. However, we ensure this is not the sole source of our performance gains:
> - **Independent Learning Capability:** As demonstrated in our ablation study (Table 3, `f + v (Randomly Init)`), even when initialized with completely random split thresholds and leaf values over the fixed topology, the differentiable ArborCell architecture stably converges and still outperforms the base XGBoost model.
> - **Input Scaling:** We preserved the native scale for tree-based models and ArbNN because their split-routing mechanisms are inherently scale-invariant. Conversely, neural baselines were normalized because they cannot be trained effectively without it. This protocol was chosen to ensure that each modeling paradigm is evaluated under its respective optimal best-practice conditions, rather than forcing an artificial constraint that would degrade the neural baselines.
>
> ------
>
> ### 4. Statistical Significance (Response to Limitations)
>
> We agree that reporting variability is crucial. To address this, we expanded our evaluation to 5 independent runs.
> *Note: Due to character limits, we report standard deviations for ArbNN and the strongest recent baselines (SwitchTab, TabPFNv2, TabM, XGBoost) below. The complete table covering all 15 baselines will be explicitly included in the revised manuscript.*
>
> | **Model**     | **MA**          | **H16**         | **CR**          | **JA**          | **DI**          | **MI**          | **CA**          | **YP**          | **HO**          | **CH**      | **ME**          |
> | ------------- | --------------- | --------------- | --------------- | --------------- | --------------- | --------------- | --------------- | --------------- | --------------- | ----------- | --------------- |
> | **SwitchTab** | 0.939±0.002     | 0.948±0.002     | 0.828±0.002     | 0.872±0.001     | 0.651±0.001     | 0.981±0.001     | 0.961±0.002     | 0.880±0.001     | 0.430±0.002     | 0.452±0.001 | 0.144±0.002     |
> | **TabPFNv2**  | 0.939±0.001     | 0.946±0.001     | 0.854±0.002     | 0.873±0.001     | 0.638±0.002     | 0.980±0.002     | 0.967±0.001     | 0.888±0.001     | 0.400±0.001     | 0.452±0.002 | 0.143±0.001     |
> | **TabM**      | 0.943±0.001     | 0.947±0.002     | 0.843±0.001     | 0.875±0.002     | 0.651±0.001     | 0.982±0.001     | 0.963±0.002     | 0.877±0.002     | 0.396±0.001     | 0.435±0.001 | 0.142±0.001     |
> | **XGBoost**   | 0.947±0.001     | 0.951±0.000     | 0.862±0.001     | 0.868±0.001     | 0.653±0.002     | 0.986±0.001     | 0.966±0.001     | 0.857±0.001     | 0.401±0.002     | 0.418±0.001 | 0.140±0.000     |
> | **ArbNN**     | **0.974±0.001** | **0.987±0.001** | **0.884±0.001** | **0.923±0.002** | **0.657±0.001** | **0.992±0.000** | **0.986±0.001** | **0.832±0.001** | **0.359±0.001** | 0.406±0.001 | **0.130±0.001** |

---

> > ### Author Rebuttal · Reviewer_8uBs · 2026-04-06
> >
> > Significant experiments are done during the rebuttal stage, which will require lot of updates to the original manuscript. I keep my score. However, I will applaud the attempt by the authors to justify the work.

---

### Decision · Program_Chairs · 2026-04-30

**Decision:**

Accept (regular)

**Comment:**

This paper proposes a new model called Arboreal Neural Networks (ArbNN) that can deal with symbolic information on tabular data but still conduct differentiable representation learning while preserving explicit rule-based reasoning. ArbNN demonstrates a structural isomorphism between a decision tree and a single-query self-attention head, which provides a differentiable architecture for logical reasoning. A notable property of the proposed approach is Bidirectional Fidelity, which enables us to initialize (compile) the model preserving the information represented by a decision tree and also turn the model back into a decision tree format. The numerical experiments show that the model is indeed effective on various public tabular data, especially on data with temporal distribution shifts. The authors also contribute to provide TabCredit, a large-scale dataset built from real-world loan applications.

The core idea of this paper, connecting differentiable architecture and explicit rule based reasoning, is interesting and actually novel. The theoretical justification that shows the structural isomorphism between a decision tree and a single-query self-attention head is elegant and offers a new perspective on the community. Furthermore, the introduction of TabCredit dataset is a valuable contribution, offering realistic benchmark for temporal distribution shifts.

The reviewers raised a concern on the reliance on a pre-trained GBDT topology. There are also some concerns on writing and formatting. The authors clarified these issues during the rebuttal phase, but these comments should be reflected to the final version of the manuscript.

Although there are some concerns as stated above, this paper provides an important and solid contribution to the community about the symbolic rule-based reasoning. Under a condition that the authors will address the issues described above, this paper is recommended for acceptance.